# Risk Assessment and Clinical Management of Children and Adolescents with Heterozygous Familial Hypercholesterolaemia. A Position Paper of the Associations of Preventive Pediatrics of Serbia, Mighty Medic and International Lipid Expert Panel

**DOI:** 10.3390/jcm10214930

**Published:** 2021-10-25

**Authors:** Bojko Bjelakovic, Claudia Stefanutti, Željko Reiner, Gerald F. Watts, Patrick Moriarty, David Marais, Kurt Widhalm, Hofit Cohen, Mariko Harada-Shiba, Maciej Banach

**Affiliations:** 1Clinic of Pediatrics, Clinical Center, Medical Faculty, University of Nis, 18000 Nis, Serbia; 2Extracorporeal Therapeutic Techniques Unit, Lipid Clinic and Atherosclerosis Prevention Centre, Immunohematology and Transfusion Medicine, Department of Molecular Medicine, “Umberto I” Hospital, “Sapienza” University of Rome, I-00161 Rome, Italy; 3Department of Internal Diseases, University Hospital Center Zagreb, 10000 Zagreb, Croatia; zeljko.reiner@kbc-zagreb.hr; 4School of Medicine, Zagreb University, 10000 Zagreb, Croatia; 5Lipid Disorders Clinic, Department of Cardiology, Royal Perth Hospital, School of Medicine, University of Western Australia, Crawley 6009, Australia; gerald.watts@uwa.edu.au; 6Department of Internal Medicine, University of Kansas Medical Center, Kansas City, MO 66104, USA; pmoriart@kumc.edu; 7Division of Chemical Pathology, Department of Pathology, University of Cape Town Health Sciences, 6.33 Falmouth Building, Anzio Rd, Observatory, Cape Town 7925, South Africa; david.marais@uct.ac.za; 8Academic Institute for Clinical Nutrition, Alserstraße 14/4, 3100 Vienna, Austria; kurt.widhalm@meduniwien.ac.at; 9Department of Gastroenterology and Hepatology, Austria Medical University of Vienna, Spitalgasse 23, 1090 Vienna, Austria; 10The Bert W. Strassburger Lipid Center, The Chaim Sheba Medical Center, Tel-Hashomer Israel, Sackler Faculty of Medicine, Tel Aviv University Israel, Tel Aviv 39040, Israel; Hofit.Cohen@sheba.health.gov.il; 11Mariko Harada-Shiba Department of Molecular Pathogenesis, National Cerebral and Cardiovascular Center Research Institute, 6-1 Kishibe-Shinmachi, Suita 564-8565, Japan; mshiba@ncvc.go.jp; 12Department of Preventive Cardiology and Lipidology, Medical University of Lodz, 93-338 Lodz, Poland; 13Department of Cardiology and Congenital Diseases in Adults, Polish Mother’s Memorial Hospital Research Institute (PMMHRI), 93-338 Lodz, Poland; 14Cardiovascular Research Centre, University of Zielona Gora, 65-038 Zielona Gora, Poland

**Keywords:** familial hypercholesterolaemia, children, cardiovascular risk, vascular phenotype

## Abstract

Heterozygous familial hypercholesterolaemia (FH) is among the most common genetic metabolic lipid disorders characterised by elevated low-density lipoprotein cholesterol (LDL-C) levels from birth and a significantly higher risk of developing premature atherosclerotic cardiovascular disease. The majority of the current pediatric guidelines for clinical management of children and adolescents with FH does not consider the impact of genetic variations as well as characteristics of vascular phenotype as assessed by recently developed non-invasive imaging techniques. We propose a combined integrated approach of cardiovascular (CV) risk assessment and clinical management of children with FH incorporating current risk assessment profile (LDL-C levels, traditional CV risk factors and familial history) with genetic and non-invasive vascular phenotyping. Based on the existing data on vascular phenotype status, this panel recommends that all children with FH and cIMT ≥0.5 mm should receive lipid lowering therapy irrespective of the presence of CV risk factors, family history and/or LDL-C levels Those children with FH and cIMT ≥0.4 mm should be carefully monitored to initiate lipid lowering management in the most suitable time. Likewise, all genetically confirmed children with FH and LDL-C levels ≥4.1 mmol/L (160 mg/dL), should be treated with lifestyle changes and LLT irrespective of the cIMT, presence of additional RF or family history of CHD.

## 1. Introduction

Familial hypercholesterolaemia (FH) is a genetic and complex multifactorial lipid disorder, which increases the risk of premature atherosclerosis and coronary artery disease [1,2,3,4]. FH is still underdiagnosed and undertreated globally and clinical strategies for the treatment and management of pediatric patients with this disorder are still far from being optimal. The objective of this position paper is to: (a) review the current approach and scientific background of cardiovascular (CV) risk stratification of children with FH, (b) to analyze available data on the clinical usefulness of other non-traditional cardiovascular risk factors as well as non-invasive methods of vascular phenotyping in children, and (c) to suggest some meaningful clinical recommendations on the potential integration of these data to help clinical-decision making and treatment planning of children with FH. 

The Position Paper was written by 10 expert representatives of the three scientific societies (Associations of Preventive Pediatrics of Serbia, Mighty Medic, and the International Lipid Expert Panel.) The level of evidence and the strength of recommendation were weighed up and graded according to predefined scales as outlined in Table 1.

Epidemiological, preventive and diagnostic aspects of Heterozygous Familial Hypercholesterolaemia in childrenHeterozygous familial hypercholesterolaemia (HeFH) is among the most common genetic metabolic lipid disorders, affecting approximately 1:200 to 1:500 of the population (1:311–1:313 based on the most recent meta-analyses) [1,2,3,4]. HeFH is characterized by elevated low-density lipoprotein cholesterol (LDL-C) levels from birth and a significantly higher risk of developing premature atherosclerotic cardiovascular disease (ASCVD) compared with subjects without FH [5,6,7,8]. However, FH is still highly underdiagnosed and undertreated, particularly in pediatric patients, and systematic preventive and clinical strategies to manage them effectively are far from optimal. To avoid overlooking children with FH and negative family history as well as the decrease in LDL-C during puberty, the National Heart, Lung, and Blood Institute and the National Lipid Association Expert Panel recently proposed universal screening as a preferred method of pediatric screening for hypercholesterolaemia between the ages of 9 to 11 years of age [9,10]. Another important paper endorsed by the European Expert Panel suggested the universal screening for children aged 1–9-years old [11]. To give a resume, both Panels recommend universal screening for hypercholesterolaemia before puberty and after one year of age. Of note, if a genetic defect has already been identified in the affected parent, an LDL-C level >135 mg/dL (3.5 mmol/L) can be used as a cut-off value for the diagnosis of FH [12].

## 2. Overall CV Risk on a Populational Level

The majority of published studies that have examined long-term cardiovascular outcomes (CV) in FH patients rely on the Simon Broome registry which has biases and limitations [13,14]. According to Copenhagen General Population Study, which is based on predefined quality parameters, with a sample of 69,016 individuals, the odds ratio (OR) for non-fatal CVD events among statin-treated and not treated FH patients was 10.3 (95%CI, 7.8–13.8) and 13.2 (95%CI, 10.0–17.4) respectively [15]. Unfortunately, this study reports an average CV risk for FH versus non-FH patients at a population level and no other additional data on the relationship between LDL-C levels and long-term CV risk within different HeFH subgroups were available. 

In clinical practice, there is a variation in ASCVD risk within FH subgroups, such as different ages when the diagnosis was established, gender, LDL-C levels, and pattern, Lp(a) levels, ethnicity, intrinsic susceptibility, genetic mutation type, treatment compliance or duration, presence of additional cardiovascular (CV) risk factors, lifestyle, etc. all of which might be important for individual patient management [7,16]. 

Recently published Korean observational study of 502,966 patients who were followed up for 14.6-years, reported that the association of FH phenotype (MEDPED criteria) and cardiovascular (CV) mortality is much smaller, with hazard ratios of 1.74 (95% confidence intervals, 95%CI: 0.96–3.15) for original MEDPED criteria and 2.18 (95%CI: 1.51–3.14) for modified MEDPED criteria [17]. Similar studies are not available from other East Asia countries, and it is hard to draw definite conclusion as to whether the CV risk in Korean patients with FH is lower due to superior genetic background or healthier different lifestyles (more fish, rice, red yeast rice in their diet, etc.) [18]. In this regard, the available data from the World Health Organization (WHO) show that East Asian countries have lower coronary heart disease (CHD) mortality than Western countries [14].

Finally, two recent updates on CV mortality in FH patients based on the Simon Broome FH register data published in 2018 demonstrated that coronary heart disease (CHD) mortality in women with FH is unchanged despite the statin treatment [19]. The most important finding was that after adjustment for traditional risk factors, the hazard ratio for CHD mortality in severe FH patients, as defined with LDL-cholesterol (LDL-C) > 10 mmol/L, or LDLC > 8.0mmol/L plus one high-risk factor, or LDLC > 5mmol/L plus two high-risk factors, was the same in all 3 subgroups, thus emphasising the importance of better clinical risk stratification, patient selection and therapeutic choices in FH patients with other risk factors [20]. 

As for gender differences in children with FH, it was recently shown that FH girls have higher levels of TC, LDL-C and non-HDL-C levels than boys from birth up to 19 years of age which may lead to their increased CV risk [21]. Of note D’Erasmo [22] et al. found that FH girls have a 2.75-fold higher risk of incident atherosclerotic cardiovascular disease than FH males (Incidence rates [IRs] 121.8 vs 33.9 per 10,000 person-years [22]. Likewise, a Norwegian registry-based study of 4688 male and female patients with a genetically confirmed diagnosis of FH, reported that CVD mortality was significantly higher in women than men with standardised mortality ratio of 3.03, 95%CI 1.76 to 5.21 in women and standardised mortality ratio of 2.00, 95%CI 1.32 to 3.04 in men [23]. 

***Expert opinion:*** It is evident that beyond LDL-C levels, other CVD risk modifiers including ethnic origin and gender contribute and/or modify overall CV risk in patients with FH (**Level B evidence**).

## 3. Lipoprotein Classes and CV Risk in FH Patients

According to the prevailing view, CV events would occur earlier in those patients with higher LDL-C levels than in those with lower LDL-C levels. However, at least four studies of untreated FH patients with clinically manifested CVD, reported no significant differences in LDL-C levels or age between study groups [6,24,25,26,27]. A 2019 study demonstrated that LDL subclass B (characterized by a predominance of small, dense low-density lipoproteins (sdLDL) has the most damaging effect on endothelial function changing the (NO)/(ONOO−) balance and contributing to the development of atherosclerosis [28]. Some recent clinical trials show that combination therapy with a beneficial effect on LDL subclass distribution is superior to LDL-C lowering alone regarding clinical events, vascular benefits, and mortality in the general population [29,30,31,32]. Of note, it was demonstrated that the genetically confirmed (GC) children with FH have unfavorable lipid profiles characterised by increased ApoB/ApoA ratio or sdLDL predominance in comparison to the non-GC FH patients [33,34]. It is also reported that low HDL-C levels and high TG/HDL-C ratio (proposed as a surrogate marker of the number of LDL-C particles) are strongly associated with a risk of CHD in patients with FH [35,36]. It is also interesting to note that increased TG/HDL ratio, apo B, apo A1 are demonstrated to be independent predictors of cIMT in children with FH [37]. 

As for the other CV risk factors, epidemiological evidence indicates a continuous association between Lp(a) levels and CV risk, with a steeper risk curve when both Lp(a) and LDL-C are elevated [38,39]. Alonso et al. suggest that the risk of CVD is highest in adult FH patients with Lp(a) level above 50 mg/dL and LDL receptor-negative mutations, while Sun et al. recently suggested that the Lp(a) level was associated with the presence and severity of CHD but not with carotid atherosclerosis in patients with HeFH [40,41,42,43]. 

So far, only one study in children found that higher levels of Lp(a) are associated with a positive family history of CVD [41]. All these data suggest a need for routine Lp(a) measurement to identify a population of high-risk children with FH who could benefit from more aggressive therapy. Most recent recommendations suggest initiating additional aggressive LDL-C lowering in FH patients with Lp(a) >50 mg/dL or even lower levels of >30 mg/dL [44,45,46,47,48,49]. From the clinical practice point of view, all these possibilities should be carefully considered when deciding to prescribe lipid-lowering treatment (LLT) in children with FH. 

The majority of the current pediatric guidelines advocate the initiation of the therapy with statins in children with FH as early as the age 8 or 10, based on the following criteria: (a) LDL-C levels > 190 mg/dL, (5 mmol/L) (b) LDL-C levels > 160 mg/dL (4 mmol/L) in presence of the family history of hypercholesterolaemia and/or premature CVD. (c) LDL-C levels > 135 mg/dL (3.5 mmol/L) in FH relatives [12,50]. 

It has to be noted that some authors allow that initiation of statins in children with FH might be at a later age, ideally started before age of 18 years (level of evidence 2—good quality clinical or observational studies) while the others suggest the treatment initiation might be needed earlier i.e., at the 6 years [51,52,53]. As for LDL-C treatment threshold, the recent consensus statement by joint working group by Japan Pediatric Society and Japan Atherosclerosis Society for Pediatric Familial Hypercholesterolaemia (FH) advocate less aggressive treatment, with statins to be considered if the LDL-C level is persistently above 180 mg/dL in children ≥ 10 years of age [54]. Descamps et al. suggest that the pharmacological treatment, using statins should not start before the age of 18 years if the LDL-C levels are <4 mmol/L (160 mg/dL) in the absence of other risk factors (hypertension, obesity, metabolic syndrome, smoking) [55]. However, this approach is against the approach that the earlier the better for the LDL-C targets, and with the well-evidenced approach that when we start earlier with FH treatment, the life expectancy, as well as the risk of CVD event, is similar to in those without a disease [53]. Of course, children with homozygous hypercholesterolaemia are by definition at extremely high CV risk and their treatment should be started immediately [56].

***Expert opinion:*** Given the above, “non-LDL-C
lipoprotein classes” and parameters might significantly modify CVD risk,
particularly Lp(a) level, and should be taken into account in clinical
decision making and CV risk stratification of FH children (**Level B evidence**).

## 4. Genotype and CV Risk

Recent genetic studies on the FH patients demonstrated poor genotype-phenotype correlation in families with the same LDL receptor (LDLR) gene defect [57,58,59]. 

Paquette et al. recently found that LDL-C levels in FH patients with the genetically confirmed mutation have no independent predictive value for CVD and that FH patients with multiple CV risk factors have a 10.3-fold higher risk for acquiring cardiovascular disease compared to FH patients with fewer CV risk factors [60]. 

The most studies on CV risk within FH families show that positive family history for premature cardiovascular disease in the first degree or second-degree relatives generally put the patient at higher risk for long-term adverse cardiovascular outcome [27,61,62]. As for children, Wiegman et al. found that children with FH and LDL-C > 6.23 mmol/L (233 mg/dL) had a 1.7-fold higher incidence (95%CI, 1.24 to 2.36) of having a parent with FH suffering from premature CVD, as well as that those children with less severe hypercholesterolaemia are unlikely to have a positive family history of premature CVD [63]. Recently Khera et al. reported that adult patients with LDL-C (>5 mmol/L) and no detected mutation have a 6-fold increase in CVD risk, and those with a known mutation have a 22-fold higher risk for CVD [26]. As of the type of LDLR mutations, it will be of particular importance to identify children with LDLR receptor-negative vs. LDLR receptor-defective mutations given that children with LDLR receptor-negative mutations had a more severe lipid phenotype (higher TC, LDL-C, and Apo B levels), higher cIMT values, than children with receptor-defective mutations [64].

Of note, Sharifi et al. found that patients with monogenic FH have greater carotid cIMT and coronary calcium score (CCS) when compared to those with polygenic hypercholesterolaemia [65]. 

Despite all these data on the very important role of the significance of genetic diagnosis in patients with FH, the therapy is still based on the phenotypic diagnosis [66,67].

***Expert opinion:*** It is reasonable to extrapolate findings from the adult population and to identify children with genetically confirmed FH with LDL-C > 5 mmol/L who are at greater cumulative lifetime exposure to high LDL-C to decide how aggressive and how early should we start with LLT (**Level C evidence**).

## 5. Traditional Risk Factors and CV Risk in FH Patients

Besides the duration of elevated LDL-C levels (cholesterol-year risk score concept in FH patients), the relationship between LDL-C levels and CV events is also dependent on some other nonlipid risk factors such as age, sex, body mass index (BMI), lifestyle factors, sympathetic nervous system, intrinsic individual differences (which might be detected in the future with genomic analysis) as well as many other more or less well known traditional and non-traditional risk factors [7,68,69,70].

In this context it is important to mention the results of Sachdeva et al. on lipid levels in 136,905 hospitalised patients with CHD [71]. They found that 77% of them had LDL-C values below 130 mg/dL (3.4 mmol/L) and almost half of them had admission LDL-C levels <100 mg/dL (2.6 mmol/L) [68]. Even though these data do not reflect the characteristics of the entire FH patient (LDL-C levels and CV risk), it can be concluded that on a population level, besides LDL-C levels, many other, more or less defined risk factors are playing an important role in CHD [72,73]. 

Obesity with all its metabolic consequences is a well-established player in atherosclerosis and other target organ damages. It is noteworthy that no significant association has been observed between body weight and an LDL-C level of ≥140 mg/dL (3.6 mmol/L) implying that if a child has an LDL-C level above this threshold and is obese, FH should be suspected irrespective of obesity [74]

Humphries et al. reported that BMI > 30kg/m^2^ significantly increased CV risk in FH patients [72]. Already more than 20 years ago Gidding et al. found significant coronary calcium in (7/29) 24% of 11 to 23 years old patients with HeFH with an increased likelihood of calcium being present in overweight HeFH patients [75]. Although it is assumed that the children with FH generally do not have problems with body mass excess, 14.1% of genetically confirmed FH (GC-FH) patients in a study of Medeiros et al. had BMI > 95th percentile and 33.8% of non-GC-FH children [34]. Of note, Kusters et al. speculate that the small difference in the annual progression of carotid intima-media thickness (cIMT) in children with FH vs their unaffected siblings 0.00041 mm/year vs. 0.0032 mm/year) is probably due to the increased prevalence of childhood obesity during past decades [76]. 

According to the SAFEHEART registry, which is based on adult patients, the annual CV event rate in FH patients is around 1% and it increases in the presence of additional risk factors. Age, male sex, history of atherosclerotic CVD before enrollment, high blood pressure, increased BMI, active smoking, LDL-C and Lp(a) levels were independent predictors of incident CVD during the follow-up [77,78]

***Expert opinion:*** It would be important to make a conceptual shift of CV risk assessment in children with FH from that based upon LDL-C levels alone, to that of combined CV risk assessment incorporating other traditional or non-traditional CHD risk factors and their possible additive adverse effects on vascular phenotype (**Level C evidence**).

## 6. Preclinical Vascular Assessment 

Elevated LDL-C levels are not the only risk factor for CV events but also for functional and structural abnormalities of the arteries which are a necessary preconditions for developing CV events. In this regard, a better characterization of the individual vascular phenotype and definition of subclinical atherosclerosis, may serve as a starting point to distinguish FH children for early intervention [78]. Only a few guidelines stress the value of non-invasive imaging of atherosclerosis in assessing and managing asymptomatic FH adults at intermediate and high risk, but none of them include FH children [79,80,81]. 

We have critically reviewed the currently available data on the clinical usefulness of the most commonly used methods to evaluate vascular and endothelial health as well as derived non-invasive surrogate atherosclerotic markers in FH children to improve clinical guidance for risk assessment and appropriate treatment planning of these children. 

## 7. Phenotypic Characterization of Children HeFH Patients

As mentioned previously, vascular phenotypes of large and small vessels may provide a new insight for studying early subclinical atherosclerosis [72,82]. Although the hard CV outcome data for HeFH children, with or without surrogate CV markers as defined by noninvasive methods, are still unavailable, several vascular phenotype parameters have already been studied to identify FH children with increased CV risk. In the recent meta-analysis the authors found that ultrasonographic measurements of cIMT and PWV (by oscillometry or applanation tonometry) are highly reproducible methods, applicable for both research and clinical practice with proven applicability for children aged ≥6 years or ≥120 cm of height, and useful for the detection of subclinical arterial damage [83].

## 8. Carotid Intima-Media Thickness (cIMT)

Studies using carotid ultrasound in healthy children show that the cIMT does not vary with age, gender, and body habitus during the pediatric age and that there is a close correlation between ultrasound and quantitative histological measurement of the cIMT during autopsy (in average 4% difference) [84]. 

Several clinical trials showed that the cIMT changes are associated with the changes in the LDL-C levels on a population level and could be used in the evaluation of the carotid atherosclerosis status [85,86,87]. 

Reported values for cIMT in a healthy pediatric population vary from 0.42 mm to 0.64 mm [87]. The most comprehensive study involving more than 1100 children from 6 to 17 years of age reported a cIMT between 0.36 mm (50th percentile at the age of 6) and 0.40 mm (50th percentile at the age of 18) using the caliper-method with the manual tracing of the contours [84]. The largest study with more than 24,000 individuals including adolescents of 15 years and older, showed that the 75 percentile for the cIMT at an age of 15 years is 0.449 mm [88]. According to the Mannheim Consensus, the 75th percentile is to be considered as the cut-off value for normal versus increased cIMT [89]. The cIMT measurement should follow the recommendations and practical guidelines for the setting, scanning, measurement and interpretation of IMT values given by the Association for European Paediatric Cardiology (AEPC) Working Group on Cardiovascular Prevention [88].

A meta-analysis by Narverud et al. of the articles presenting data on cIMT revealed significantly thicker cIMT in children with FH compared with controls thus strengthening the evidence of early atherosclerotic development in children with FH [90]. Likewise, a recent meta-analysis in the adult population with FH also showed that cIMT is increased when compared with non-FH adult controls [68].

Braamskamp et al. showed that increased cIMT in statin-treated children with FH decreases during 6–12 months while Bos et al. showed long-term statin treatment in HeFH patients reduces carotid atherosclerosis to a degree of the healthy population [91,92]. 

However, the existing data on this are still inconsistent; a 10-year follow-up study in statin-treated children with FH and their unaffected siblings showed that the mean cIMT was significantly greater in children with FH even after 10 years of treatment with lipid-lowering medication although the progression of the cIMT from baseline remained similar in both groups. Likewise, regression or slowed progression of cIMT in adults induced by cardiovascular drug therapies was not reflected in the reduction of cardiovascular events [93]. 

However, one should have in mind, that all those results might be an effect of the time when lipid-lowering therapy was introduced (the earlier the better), the baseline changes of cIMT, as well as of intensity of therapy. 

As for the usefulness of cIMT measurement alone to predict CV events, the results of meta-analyses are conflicting [93]. However, it is worth noting that no meta-analysis on this issue took into consideration the heterogenicity of the analysed population with regard to their different long-term CV risk profiles. The fact that many CV hard outcomes in previous cohorts studied by meta-analyses certainly occurred in FH patients, which are by definition at the highest CV risk, make the generalization of their conclusions less accurate. Recently Dyrbuś et al. found that in adult Polish patients with a history of acute coronary syndrome almost 1.6% had probable/definitive FH (4 times more than in the whole population) and 17% had possible FH [94]. Hence, it would be important to further refine CV risk estimation in FH patients to have data on the predictive value of cIMT in terms of CV risk in children as well as in adults with FH.

One of the key methodological issues is that, even in high-risk populations, both in children and in adults, changes in cIMT over time are too small to be captured with ultrasound cIMT scans, even when measurements are repeated after several years. It was shown that the annual rate of cIMT progression in children with FH is 0.00041 mm/year, and therefore below the resolution of carotid ultrasound (~0.3 mm) [76]. Therefore, whether the dynamic cIMT changes reflect a true change in risk of future CVD events in FH children has still to be proven. This is an important call for action to find innovative and more accurate measurements to monitor atherosclerosis progression. One that should be further discussed is angio-computed tomography and calcium scoring measurement. The recently introduced hypothesis of the “power of zero”, as well as the results >1 and especially >100 might be a very good tool both for the prediction as well as for the optimal treatment introduction [95,96,97].

***Expert opinion***: Although longitudinal data on the association between cIMT in children with FH and hard CV outcomes are still lacking and although children with FH cannot be distinguished individually based on their cIMT diameters it would be useful to consider the nearest one decimal value to 75th percentile of normal cIMT in children (0.5 mm) as a threshold for treatment initiation of FH children. FH children with cIMT ≥0.4 mm should be carefully monitored to initiate lipid-lowering management at the most suitable time (**Level B evidence**).

## 9. Endothelial Dysfunction

Endothelial dysfunction is an integrated index of both, the global CV risk-factor burden and the sum of all vasculoprotective factors in an individual [98]. It is considered a key event in the initiation, progression, and complications of atherosclerosis. Lipids, particularly LDL-C, play the most important role in endothelial dysfunction by reducing the bioavailability of nitric oxide (NO) and activating proinflammatory signaling pathways [99].

A systematic review and meta-analysis by Masoura et al. with 4057 FH patients (both adults and children) demonstrated that the severity of hypercholesterolaemia was associated with the presence of arterial function impairment as assessed by brachial artery flow-mediated brachial dilation (FMD), which is the most common method for noninvasive assessment of the endothelial function in children [100,101]. However, there is still a lot of inconsistent results of such analyses in children. At least 6 studies on the clinical utility of FMD in children with FH showed no correlation between lipid levels and FMD [49,101,102,103,104].

Lewandowski et al. and Järvisalo et al. found that FMD response is inversely associated with serum cholesterol concentrations, while on the other hand, Deanfield et al. found no association between HDL and LDL levels and endothelial function [49,105]. In two more recent studies, FMD was found to be significantly decreased in children with FH aged >10 years when, compared to control subjects [106,107,108].

To date there are two studies providing reference values of FMD in a group of healthy children with enough statistical power (a minimum of 200 children required) [109]. Although there were no significant technical and methodological differences between both studies the reference FMD values significantly differed in-between both studies, i.e., FMD max in a group of 13-year-old male children, was 9.5 ± 4.3 (boys) in the first study and 7.86 ± 0.85 in the second study for both boys and girls? [110]. 

Besides wide reference limits, another barrier for individual vascular phenotype assessment by FMD operator dependence and wide variation in brachial artery response during the day. 

Apart from FMD, at present some novel methods for in vivo endothelial function assessment, including digital thermal monitoring (DTM), venous occlusion plethysmography (VOP), are introduced into clinical settings for research purposes [111,112]. However, none of these methods has been currently applied in the pediatric population due to the lack of technique and methods standardisation.

## 10. Arterial Stiffness

Vascular stiffness is another indicator of arterial health, which is dependent on vascular structure, function, and arterial pressure. It can be quantitated by analysis of arterial pressure waveforms, changes in diameter (or area) of an artery with respect to the distending pressure and by assessing the velocity of pulse-wave travel (PWV). Increasing evidence suggests that aortic stiffness measured by PWV could be a reliable biomarker that integrates, in a single measurement, the overall burden of CV risk factors on the vasculature during the over time [113]. However, it has not been proven so far whether measures of arterial stiffness can be used as a surrogates for atherosclerotic disease as well as for monitoring the efficacy of CVD treatment in children with FH. 

## 11. Pulse Wave Velocity (PWV)

PWV has emerged as an important parameter, for the measurement of arterial stiffness and is considered a useful surrogate marker in assessing atherosclerotic development and CV risk, in adults with CV risk as well FH patients [114,115,116]. There have been many clinical studies and meta-analyses in adults showing the association between PWV and coronary/cerebral/carotid atherosclerosis in the adult population [117]. A meta-analysis of prospective observational data from 17,635 adult subjects from 17 cohorts showed that the addition of PWV improved the CVD risk prediction, especially in intermediate-risk and younger individuals [118]. 

Important limitations for PWV implementation in pediatric clinical work were different methodological approaches for PWV measuring, as well as the lack of reference values for children [107,119,120,121,122]. Nevertheless, given the low likelihood of validation studies to be performed in pediatric FH patients it became more important to assess precision and reproducibility than accuracy (validity) when attempting to analyse the vascular phenotype in FH children [123,124].

Riggio et al. were the first to suggest that PWV, automatically calculated by the echo-tracking method, and augmentation index but not carotid intima-media thickness, are early indicators of vascular damage in hypercholesterolemic children [125]. Aggoun et al. also found increased stiffness of the common carotid artery in children with FH independently of blood pressure levels [126]. In the largest study of 267 adolescents, PWV was significantly elevated in those with high LDL-C [127]. Recently Tran et al. reported that the PWV as assessed by cardiac MRI is significantly increased (*p* < 0.001) in children with FH when compared to age- and sex-matched reference data [128]. Opposite to these results, Vlahos et al. found no difference in central pulse wave velocity in a group of 30 children with FH, measured noninvasively using applanation tonometry technique. However, this study had many methodological limitations [107].

Recently, reference values for the PWV in healthy children have been established [129,130,131,132,133,134]. They provide the largest database concerning Ao-PWV in healthy children and adolescents and may be of additional value to improve diagnostics and risk stratification of children with FH.

A study by Reusz et al. published in 2010 performed in a cohort of >1000 children and teenagers aged between 6 and 20 years was the first that provided sex-specific reference curves for age and height and distribution for PWV, using applanation tonometry measurement [129]. However, the wide reference range in this study is the result of practical problems measuring PWV in younger children since up to one-quarter of carotid-femoral tonometry data could not be acquired due to difficulties to make them sit still, to palpate pulses or to obtain traces of adequate quality, which all limits its clinical benefit. Of four subsequent studies, two studies conducted on healthy children in Latin America and Europe were based on oscillometric technique with Arteriograph device (requires external measurement of the jugulum symphysis distance to calculates the timing of the brachial wave reflection); one study used Vicorder device (automatically marks the pulse wave’s steepest ascending part and uses a defined timeframe to detect the wave’s nadir to calculate transit time.) and one study used Mobil-O-Graph device (analyse pulse wave and wave separation using the inbuilt ARCSolver algorithm) [131,132,133,134]. 

It should be noted that after applying a path length adjustment for the oscillometric and applanation technique both methods with all mentioned different devices provided comparable results. Shortened synopsis of PWV 95 and 97.5th percentiles (equal to 2 standard deviations) of pediatric PWV normative data according to age and sex obtained with oscillometric devices (Arteriograph - Hidvegi et al); (Vicorder - Thorn et al) and using applanation technique (PulsePen device-Reusz et al) is presented in Table 2. The other two studies haven’t provided tabular values of PWV percentile categories making them less practical for clinical usage. Considering the methodological issue, the Arteriograph uses one cuff but needs the external measurement of the jugulum-symphysis distance, while Vicorder device uses two cuffs, neck and femoral which could be possible practical clinical limitations for routine usage. In general, Mobil-O-Graph has a little advantage over other devices in terms that direct palpation of the artery is not required (also not required for Vicorder device), and pulse travel distance measurement is not necessary which is more important. It is important to underline that only a small error in measurement of path length can influence the absolute value of pulse wave velocity and can lead to enormous variations in measurements [135]. Of note, inter- and intraobserver variability of measurements, obtained with Mobil-O-Graph device, is of good reproducibility inter- and intraobserver variability [136]. Also, unlike the Vicorder device, Mobile-O-Graph requires only one site of pressure waveform recording. In general, all devices mentioned above provide slightly different measures of PWV and their reference values should be used separately unless corrected for path length.
***Expert opinion:*** Depending on the availability of noninvasive equipment for PWV measurement and staff experience, it would be clinically meaningful to perform PWV measurements (preferably via oscillometric device we suggest the usage of Mobile-O-Graph device due to the simplicity of measurement) in all children with FH and evaluate their changes over time. PWV values above 97th (See Table 2) could be a possible guide for treatment initiation in ambiguous clinical cases (**Level B evidence**).

## 12. Novel Biochemical Biomarkers and CV Risk in FH Children

In recent years an increasing number of studies have been published on the usefulness of novel biochemical cardiovascular biomarkers (BM) in the risk stratification of children with FH. However, the small sample size and cross-sectional design limit the generalisability of their results and do not allow establishing prognostic utility. From a clinical perspective endothelial dysfunction marker plays a main role in the atherosclerotic process. The binding of circulating leukocytes to the vascular endothelium by their interaction with cell adhesion molecules is considered a crucial step leading to the initial recruitment of leukocytes into the vascular wall, low-grade systemic inflammation and atherogenesis [137]. For this reason, P-selectin, E-selectin, I-CAM-1, V-CAM-1, von Willebrand factor, thrombomodulin, plasminogen activator inhibitor-1 (PAI-1) plasma levels, hs-CRP and PAI-1/tPA ratio are the most investigated BM measuring EC dysfunction in FH patients. To date there are only a few studies demonstrating an association between endothelial dysfunction markers and increased vascular risk in FH children [138,139]. Increased intercellular cell adhesion molecule (ICAM-1) in FH children was reported in one study and an association between P-selectin levels and carotid IMT was reported in another study [140,141,142]. As for hs-CRP, data from 11 studies are discrepant, implying that circulating levels of CRP may be a less sensitive marker of atherosclerotic development in children with FH [90]. Some other studies reported higher levels of other inflammatory or mediators, including, interleukin-6 (IL-6), tumor necrosis factor (TNF)- α in FH children compared to controls [102]. A novel source for plasma-derived markers related to increased CV risk in FH patients, includes reactive oxygen species (ROS), as well as some novel biomarkers that have been identified through highly sensitive proteomic techniques.

***Expert opinion:*** Further studies specifically addressing new biochemical cardiovascular biomarkers in FH children are warranted since the correlation between them and serum cholesterol were not found consistently. Whether the measurement of these biomarkers might also contribute to CVD risk stratification in FH children needs further evaluation. 

## 13. Treatment and Phenotype Characterisation

Pharmacological treatment of children with FH is the most challenging task for pediatricians and has not changed much in recent decades. The Cochrane systematic review from 2014 indicated that dietary interventions recommended for FH are not proven to prevent CHD, and statins remain the basis of medical management for most FH patients [143]. However, this review does not address the effect of diet on lipoprotein levels in FH-children although a few studies show that diet is able to lower LDL-C levels in the range of 10–15% [144]. Lipoprotein apheresis and relatively recently approved lipid-lowering drugs such as the PCSK9 inhibitors are in most countries available but are not widely accessible for children and we are still waiting for the final results of phase 3 and CVOT studies with these drugs. Also cost-effectiveness data as well as long term safety data are lacking for PCSK9 inhibitors in this group of patients and the answer to these questions will probably be provided by the HAUSER-RCT study which is an ongoing, phase 3, randomised, placebo-controlled, double-blind, parallel-group, multicenter study designed to assess the efficacy, safety, and tolerability of evolocumab in pediatric patients aged 10–17 years with HeFH, and The ODYSSEY KIDS study, phase 2, randomised, placebo-controlled, double-blind, parallel-group, multicenter study designed to assess the efficacy, safety, and tolerability of alirocumab [145,146,147,148]. Given the results of Humphries et al study, that patients with the PCSK9 mutation have the highest CHD risk, the children, and adolescents with such a mutation (although extremely rare) may be the best candidates for these drugs [138]. A recent 24-week, randomised, double-blind, placebo-controlled trial of evolocumab in pediatric patients with heterozygous familial hypercholesterolaemia showed its excellent LDL-C–lowering efficacy, tolerance, and safety [139]. 

As for statins, it is still difficult to decide how aggressive and how early should we start to prevent premature atherosclerosis as well as how to monitor the effects of this treatment [149]. The main indication for these drugs and their dosage is still based on arbitrary LDL-C targets and we still do not consider vascular phenotype status and other risk factors before the treatment initiation. There are also some new promising approaches for the treatment of patients with FH as regard gene- and cell-based therapies but no experience with FH children exists so far [150].
We recommend measurements of cIMT and PWV in all children with FH as a routine CV phenotype and risk assessment procedure. It would be rational to accept “wait and see approach” in children with both genetically confirmed or non-confirmed FH with LDL-C levels between 130–160 mg/dL (3.5–4.1 mmol/L) and no existing structural subclinical vascular changes as detected by carotid ultrasound (normal carotid intima-media thickness—cIMT) measurement. See the proposed algorithm for clinical evaluation and treatment of children and adolescents with HeFH assuming that before the treatment initiation, other risk factors (history of statin intolerance, thyroid functions, liver or kidney disease; etc.) also must be evaluated Table 3 (**Level C evidence**).


**Table 3 jcm-10-04930-t003:** The summary of the clinical practice recommendation of the ILEP, MM and APPS.

LDL-C Values	cIMT	Risk Factors	Genetic Confirmation	Treatment
3.5–4.1 mmoL/L	⊥	(−)	(−)	(LM *)
3.5–4.1 mmoL/L	↑	(±)	(±)	LM * + LLT **
3.5–4.1 mmoL/L	⊥	(±)	(+)	LM *
4.1–5.0 mmoL/L	⊥	(−)	(−)	LM *
4.1–5.0 mmoL/L	⊥	(+)	(−)	LM * + LLT **
≥4.1 mmoL/L	⊥/↑	(±)	(+)	LM * + LLT **
≥5.0 mmoL/L	⊥/↑	(±)	(±)	LM * + LLT **

ILEP—International Lipid Expert Panel (ILEP), MM—Mighty Medic and APPS—Association of Preventive Pediatrics of Serbia; ⊥—normal; ↑—increased; PWV—Pulse Wave Velocity; cIMT—Carotid intima media thickness LM * Lifestyle modifications ** Start LLT (statins and/or ezetimibe) between 8–10 years of age with target LDL-C < 3.5 mmol/L (130 mg/dL) if >10 years, or ideally 50% reduction from baseline if 8–10 years.

## 14. Conclusions

It may be reasonable to start phenotypic vascular assessment (cIMT measurement and PWV) at the age of 8 years when the first significant structural differences in cIMT between FH and non-FH children were described.

Depending on the availability of non-invasive equipment and staff experience, cIMT measurement with or without PWV measurement (preferably via Mobil-O-Graph oscillometric device) may be considered to characterise the vascular phenotype.

As a structural vascular phenotype marker, a cIMT of 0.5 mm should be used as a first threshold for treatment initiation in FH children, however still values ≥0.4 mm should be treated as a risk and carefully monitored. 

Abnormal PWV threshold values > 97 percentile (Table 2), obtained via oscillometric technique, could be a possible guide for treatment initiation in ambiguous clinical cases.

Nonpharmacological well-adherent lifestyle modification (low-fat diet enriched with soy protein and physical activity) should be introduced in all low-risk FH children, (non-genetically confirmed, negative family history for premature CV disease, absence of traditional or non-traditional CV risk factors) who do not have severe hypercholesterolaemia (LDL-C between 3.5–4.1 mmol/L (130–160 mg/dL).

All FH children with LDL-C levels between 3.5–4.1 mmol/L (130–160 mg/dL) and abnormal cIMT thickness should be treated with LLT along with LM preferably between 8–10 years.

Genetically confirmed children with FH and LDL-C levels between 3.5–4.1 mmol/L (130–160 mg/dL) and normal cIMT thickness should be treated with LM irrespective of the presence of additional RF or family history of CHD. All genetically non-confirmed children with FH and LDL-C levels 4.1–5 mmol/L (160–190 mg/dL) and normal cIMT should be closely monitored for the occurrence of both structural and functional vascular abnormalities (PWV), and additional comorbid conditions. Nonpharmacological LM should be immediately introduced and monitored. LLT treatment should be introduced if any of the RF is present.

We propose LLT together with LM in all children with genetically confirmed FH and LDL-C levels 4.1–5 mmol/L (160 mg/dL) as well as those children with FH and LDL-C levels ≥5.0 mmol/L, irrespective of the cIMT, presence of CV risk factors and positive familial history, preferably between 8–10 years. In those with very high LDL-C levels and subclinical vascular changes one should consider PCSK9 inhibitors on top of statins and ezetimibe, as only phase 3 studies will confirm their efficacy and safety and the extension of indications will be officially approved. 

Further longitudinal studies to evaluate dynamic changes of the characteristics of vascular phenotype intermediate endpoints (including endothelial function) during LLT treatment will further contribute to the better understanding of the development of atherosclerosis in children with FH as well as their better and more personalised clinical management.

## Figures and Tables

**Table 1 jcm-10-04930-t001:** Classification of the level of evidence.

Level of Evidence	Definition
Level A	Data derived from multiple randomised clinical trials or their meta-analysis
Level B	Data derived from a single randomised clinical trial or large non-randomised studies
Level C	Consensus or opinion of experts and/or small studies, retrospective studies, registries

**Table 2 jcm-10-04930-t002:** Synopsis of recently published normative data for aortic PWV according to age for Males (
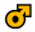
) and Females (
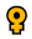
).

Years	95th 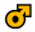 *	95th 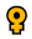 *	97th 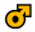 **	97th 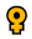 **	97th 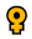 ***	97th 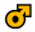 ***
8	5.451	5.400	6.71	6.76	4.62	4.52
9	5.513	5.543	6.74	6.80	4.73	4.63
10	5.615	5.684	6.79	6.87	4.83	4.76
11	5.758	5.814	6.86	6.95	4.94	4.88
12	5.919	5.918	6.97	7.02	5.03	5.02
13	6.089	6.003	7.11	7.08	5.11	5.18
14	6.271	6.093	7.22	7.11	5.16	5.36
15	6.471	6.195	7.28	7.12	5.19	5.54
16	6.675	6.316	7.31	7.11	5.20	5.70
17	6.874	6.469	7.35	7.15	5.23	5.82
18	7.082	6.654	7.44	7.26	5.28	5.93

*—Reusz et al. [129]; **—Hidvegi et al. [131] and ***—Elmenhorst et al. [133].

## Data Availability

All the data are available in the main text.

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
