# Peer review of "Risk Assessment and Clinical Management of Children and Adolescents with Heterozygous Familial Hypercholesterolaemia. A Position Paper of the Associations of Preventive Pediatrics of Serbia, Mighty Medic and International Lipid Expert Panel"

_jcm, 2021, doi:10.3390/jcm10214930_

Round 1

Reviewer 1 Report

The paper covers the important matter of risk assessment and clinical management of FH children and adolescent, an issue that needs more attention in order to reduce the burden of CV risk in FH population.

Nonetheless there are some points of criticism:

- One of the most relevant barrier to a proper risk assessment and clinical evaluation of FH children is the very low identification rate of young FH patients. A paragraph about current and desirable strategies for FH identification in children and adolescent is strongly recommended.

- Data on adult FH population are presented throughout the paper and sometimes it does not seem to deal with children and adolescent (Es. Paragraph 2). Please clarify this issue.

- It is important to underline the relevance of FH molecular diagnosis in children (Paragraph 4). In particular the concept about different phenotypes in patients carrying LDLR receptor-defective vs. LDLR receptor-negative mutations should be reported. Please modify and enrich this part of the paper.

- It is difficult to understand the sentence about LDL-C levels in CHD patients (Paragraph 5, line 183 to 189). It is well known that most of CHD patients have normal or even low LDL-C values. But this does not conflict with the concept that LDL-C is a relevant risk factor for atherosclerosis. This sentence is particularly unexpected in a paper about familial hypercholesterolemia. Please clarify.

- Paragraphs 9 to 11 seem to be sub-paragraphs of Preclinical vascular assessment (P. 6) and Phenotypic characterization of children HeFH patients (P. 7). In particular paragraphs about cIMT, Endotelial dysfunction and PWV should be shortened with the identification of relevant and practical concepts about risk stratification in children and adolescent.

- Paragraph about novel biochemical markers does not add much to “real-life” phenotypic characterization of FH patients, at least until now. Please consider removal.

In conclusion, Authors should clarify the key messages of their work: what a physician should do or avoid to fully characterize and treat young young FH patients. A final table or a summary could be useful for this purpose.

Author Response

请查看附件

Reviewer 2 Report

The paper " Risk assessement and clinical management of children and adolescents with heterozygous familial hypercholesterolemia. Are we ready for the wait and see approach" is an excellent review of the state of the art and raises interesting questions for future research and may have interest in the clinical practice. 

The research is relevant and interesting since reviewing the topic of the diagnosis and treatment of Familial Hypercholesterolemia in childhood. It is especially relevant in that it means an update in the assessment of vascular risk in children with Familial Hypercholesterolemia. The paper is well written, it is clear and easy to read.  Conclusions are consistent and answer the question of how to follow and manage familial hypercholesterolemia from an early age.

Author Response

Dear Editor,

We would like to thank the reviewer for the expert comments.

REVIEWER 2 EVALUATION

The paper " Risk assessment and clinical management of children and adolescents with heterozygous familial hypercholesterolemia. Are we ready for the wait and see approach" is an excellent review of the state of the art and raises interesting questions for future research and may have interest in the clinical practice.

The research is relevant and interesting since reviewing the topic of the diagnosis and treatment of Familial Hypercholesterolemia in childhood. It is especially relevant in that it means an update in the assessment of vascular risk in children with Familial Hypercholesterolemia. The paper is well written, it is clear and easy to read.  Conclusions are consistent and answer the question of how to follow and manage familial hypercholesterolemia from an early age.

We are pleased with your positive decision

Round 2

Reviewer 1 Report

The revised manuscript follows some of the advice reported in the revision, in particular clarification of some points and reduction of some sections.

The absence of a final table/summary represents an important limitation to the message of the paper.

Author Response

We would like to thank reviewers for their further expert comments. 

We have also amended the manuscript as suggested.

Table 2. The summary of the clinical practice recommendation of the ILEP, MM and APPS.

LDL-C values

cIMT

Risk factors

Genetic confirmation

Treatment

Non invasive screening  (PVW, cIMT)

3.5-4.1 mmoL/L

^

(-)

(-)

(-)

Every 6 months

3.5-4.1 mmoL/L

­

(±)

(±)

LLT*

Every 6 months

≥4.1 mmoL/L

^

(-)

(-)

LM*** + statins

Every 6 months

≥4.1 mmoL/L

^/­

(±)

(+)

LLT**

Every 6 months

ILEP - International Lipid Expert Panel (ILEP), MM - Mighty Medic and APPS - Association of Preventive Pediatrics of Serbia; ^ - normal; ­ - increased; PVW – Pulse Vave Welocity; cIMT - Carotid intima media thickness *Start LLT with statins and or ezetimibe between 8-10 years of age with target LDL-C < 3.5 mmol/L (130 mg/dL) if > 10 years, or ideally 50% reduction from baseline if 8–10 years; **Start LLT with statins and or ezetimibe as early as possible (preferably at 8 years of age) with target LDL-C < 3.5 mmol/L (130 mg/dL) especially with presence of subclinical vascular changes, elevated lipoprotein(a), family history of premature CHD or arterial hypertension; LM*** Lifestyle modifications.

Round 3

Reviewer 1 Report

The paper represents a good attempt to present current and future strategies for FH characterization in young patients.

In the revised form and with the final table explaining key messages of the work, the paper can be considered an important aid for the management of young FH patients.

Author Response

Thank you for your support of our work.